# Subgenome Discrimination in *Brassica* and *Raphanus* Allopolyploids Using Microsatellites

**DOI:** 10.3390/cells10092358

**Published:** 2021-09-08

**Authors:** Nicole Bon Campomayor, Nomar Espinosa Waminal, Byung Yong Kang, Thi Hong Nguyen, Soo-Seong Lee, Jin Hoe Huh, Hyun Hee Kim

**Affiliations:** 1Department of Chemistry and Life Science, BioScience Institute, Sahmyook University, Seoul 01795, Korea; nbacampomayor@syuin.ac.kr (N.B.C.); newaminal@syu.ac.kr (N.E.W.); kang1969@syu.ac.kr (B.Y.K.); nguyenhong@syuin.ac.kr (T.H.N.); 2BioBreeding Institute, Ansung 17544, Korea; sslee0872@hotmail.com; 3Department of Agriculture, Forestry and Bioresources, Plant Genomics and Breeding Institute, Seoul National University, Seoul 08826, Korea; huhjh@snu.ac.kr

**Keywords:** allopolyploid, *Brassica*, fluorescence in situ hybridization, karyotype, microsatellite, *Raphanus*

## Abstract

Intergeneric crosses between *Brassica* species and *Raphanus sativus* have produced crops with prominent shoot and root systems of *Brassica* and *R. sativus*, respectively. It is necessary to discriminate donor genomes when studying cytogenetic stability in distant crosses to identify homologous chromosome pairing, and microsatellite repeats have been used to discriminate subgenomes in allopolyploids. To identify genome-specific microsatellites, we explored the microsatellite content in three *Brassica* species (*B. rapa*, AA, *B. oleracea*, CC, and *B. nigra*, BB) and *R. sativus* (RR) genomes, and validated their genome specificity by fluorescence in situ hybridization. We identified three microsatellites showing A, C, and B/R genome specificity. ACBR_msat14 and ACBR_msat20 were detected in the A and C chromosomes, respectively, and ACBR_msat01 was detected in B and R genomes. However, we did not find a microsatellite that discriminated the B and R genomes. The localization of ACBR_msat20 in the 45S rDNA array in ×*Brassicoraphanus* 977 corroborated the association of the 45S rDNA array with genome rearrangement. Along with the rDNA and telomeric repeat probes, these microsatellites enabled the easy identification of homologous chromosomes. These data demonstrate the utility of microsatellites as probes in identifying subgenomes within closely related *Brassica* and *Raphanus* species for the analysis of genetic stability of new synthetic polyploids of these genomes.

## 1. Introduction

The genus *Brassica* (Brassicaceae) includes several economically valuable crops that are used as oilseeds, condiments, culinary vegetables [1,2], and sources of health-promoting phytochemicals [3,4]. *Brassica* is also a good model for studying polyploidization because of numerous intercrossing and morphological variants [5,6]. The genetic relationship of the six most economically important *Brassica* species was described in U’s triangle [7]. These include three diploids, *B. rapa* (AA genome, 2*n* = 2x = 20), *B. nigra* (BB, 2*n* = 2x = 16), and *B. oleracea* (CC, 2*n* = 2x = 18), and their allotetraploid hybrids, *B. juncea* (AABB, 2*n* = 4x = 36), *B. napus* (AACC, 2*n* = 4x = 38), and *B. carinata* (BBCC, 2*n* = 4x = 34).

Like many *Brassica* species, *R. sativus* L. (RR, 2*n* = 2x = 18) has also been cultivated worldwide [8,9]. *R. sativus* is valued more for its roots than *Brassica* species, whose shoots are the primary commodities. Breeders have attempted to synthesize hybrids with shoot and root features of *Brassica* crops and *R. sativus*, respectively, to maximize the use of plant parts for culinary purposes [10]. Therefore, several synthetic crops that involved crosses between the A, C, B, and R genomes have been generated [11,12,13,14,15]. One such example is the intergeneric allotetraploid ×*Brassicoraphanus*, commonly known as ‘Baemoochae’, which resulted from a cross between *B. rapa* ssp. *pekinensis* L. and *R. sativus* L. [10,16].

However, due to frequent genetic instability of the selected phenotypes, only a few stable ×*Brassicoraphanus* lines have been successfully registered as commercial cultivars to date. One of these is “BB#1′’ (2*n* = 38, AARR genome), which was stabilized through induced microspore mutation of the intergeneric hybrid between *B. rapa* cv. Weongyo#207 and *R. sativus* cv. Baekgyoung [10,17,18].

Several lines involving multiple genomes have also been developed from crosses of synthetic allotetraploids. One example is hybrid ×*Brassicoraphanus* 977 (Appendix A), which possibly carries chromosome blocks from the A, C, and R genomes, considering its genomic background from a cross between two ×*Brassicoraphanus* allotetraploids, BB#50 (AARR, 2*n* = 4x = 38) and “Mooyangchae” (RRCC, 2*n* = 4x = 36), a fertile amphidiploid progeny between *Raphanus sativus* cv. HQ-04 (2*n* = 2x = 18, RR) and *Brassica alboglabra* Bailey (2*n* = 2x = 18, CC) [19]. At present, the karyotype and genome composition of ×*Brassicoraphanus* 977 have not yet been reported.

It is important to identify the donor genomes of all chromosomes in ×*Brassicoraphanus* 977 for efficient karyotyping. Genomic in situ hybridization (GISH) is a conventional technique used to analyze donor genomes in allopolyploids [20,21]. However, GISH is laborious and inefficient in discriminating the A and C *Brassica* genomes in *B. napus* [22,23,24]. However, GISH-like results have been achieved using conventional fluorescence in situ hybridization (FISH) with microsatellite repeats as probes instead of genomic DNA. This approach discriminated C from the A genome in *B. napus* [24,25] and the B genome in *Triticum aestivum* [26].

Microsatellites are often used in PCR-based genetic diversity studies because of their high rate of transferability to other closely and even distantly related taxa [27]. They have been used to identify subgenome alleles in some bryophytes [28]. Moreover, microsatellites are good cytogenomic markers because they are expected to be distributed evenly along chromosomes [29]. However, they have not yet been used to visually discriminate subgenomes in allopolyploid species, and there has not been an extensive comparative whole-genome microsatellite quantification and FISH analysis between the *Brassica* A, C, B, and *R. sativus* R genomes. Hence, we are advancing the use of microsatellites in analyzing their presence and absence in component chromosomes of subgenomes to identify homologous chromosomes in allopolyploids.

This study aimed to determine whether similar microsatellites, such as the C-genome-specific ones found in *B. napus*, are also found in the A, B, and R genomes. We quantified microsatellite repeats in the A, C, B, and R genomes and performed comparative FISH to identify potential genome-specific microsatellites that can be used as a cytogenomic resource in karyotyping polyploid hybrids between these four genomes. Finding similar microsatellites unique to the A, C, B, and R genomes would ease chromosome discrimination in ×*Brassicoraphanus* 977 and other natural and synthetic allopolyploids carrying these genomes.

## 2. Materials and Methods

### 2.1. Microsatellite Mining

To analyze genome-specific microsatellites, we downloaded 100-bp whole-genome sequence reads of the four diploid A, C, B, and R genomes from NCBI (Table 1). High-quality reads representing 0.04–0.06× of the A, C, B, and R genomes [30,31,32,33] were selected using the FastQC tool in the RepeatExplorer version 2 pipeline [34]. Following a script provided in RepeatExplorer2, a long single contig was generated by concatenating all reads from each species and inserting 50-bp Ns between reads.

Each contig from the A, C, B, and R genomes was scanned for tandem repeats of ≤ 20 bp using Tandem Repeats Finder version 4.09 [35], and the output was sorted and quantified using the TRAP program v1 [36]. The repeat units of each microsatellite were concatenated to approximately 200 bp. To filter out redundant sequences, we grouped similar sequences and generated consensus sequences using the *Assembly Sequence* features in the CLC Main Workbench (CLC Inc., Aarhus, Denmark). The microsatellites were named ACBR_msat# according to the closer relationship of the A/C and B/R genomes [37], msat for “microsatellite” and the number order according to decreasing cumulative abundance. For FISH validation, we selected microsatellites with > 6 kb cumulative total bases from the A, C, B, and R genomes. A summary of the mining and FISH validation workflow is presented in Appendix A.

### 2.2. Designing Microsatellite Pre-Labeled Oligoprobes

We developed pre-labeled oligonucleotide probes (PLOPs) for FISH validation of candidate microsatellites (Appendix A). PLOPs were designed using the CLC Main Workbench (CLC, Inc.) and synthesized by Bio Basic Canada Inc. (Toronto, ON, Canada). In addition, we also used 5S rDNA, 45S rDNA, and telomeric repeat PLOPs for chromosome identification. The rDNA and telomeric PLOPs were designed and prepared following methods described in [38] and were purchased from Bioneer (Daejeon, Korea).

### 2.3. Plant Samples

Seeds of the four diploid and five allopolyploid species were purchased commercially and provided by the National Plant Germplasm System of the US Department of Agriculture, the National Academy of Agricultural Science of Korea, or the BioBreeding Institute, Ansung, Korea (Table 2). Seeds were germinated on filter paper and incubated at 25 °C. Harvested root tips were treated with 2 mM 8-hydroxyquinoline for 5 h at 18 °C, fixed with aceto-ethanol (1:3 *v*/*v*), and stored in 70% ethanol at 4 °C until use.

### 2.4. Chromosome Preparation

Somatic chromosome spreads were prepared following the technique described by Waminal and Kim [39]. Five meristematic tips (~2 mm) were digested with pectolytic enzyme solution (2% Cellulase R-10 (C224, Phytotechnology Laboratories, Lenexa, KS, USA) and 1% Pectolyase Y-23 (P8004.0001, Duchefa, Haarlem, The Netherlands) in 100 mM citrate buffer) at 37 °C for 90 min. Roots were transferred into a microtube with chilled Carnoy’s solution and vortexed for 30 s at room temperature. The supernatant was discarded, and the pellet was resuspended in an appropriate amount of aceto-ethanol (9:1 *v*/*v*). The cellular suspension was pipetted onto pre-cleaned slides and pre-warmed in a humid chamber. After air-drying, slides were fixed in 2% formaldehyde [39] for 5 min, quickly rinsed with distilled water, and dehydrated in a series of ethanol treatments (70%, 90%, and 100%).

### 2.5. FISH and Karyotyping

FISH was performed according to the modified procedure of Waminal et al. [40] and Lim et al. [41]. The FISH hybridization mixture was prepared with 100% formamide, 50% dextran sulfate, 20× SSC, 50 ng/μL of each DNA probe, and deionized water (Sigma-Aldrich, St. Louis, MO, USA). The mixture was pipetted onto each slide. Slides were incubated at 80 °C for 5 min and transferred to a humid chamber at 37 °C overnight. After overnight hybridization, incubated slides were washed in 2× SSC at room temperature (RT) for 10 min, 0.1× SSC at 42 °C for 25 min, and 2× SSC at RT for 5 min, and dehydrated in a series of alcohol (70%, 90%, 95%). Slides were air-dried and counterstained with 4′,6-diaminidino-2-phenylindole (DAPI) at a ratio of 1:100 DAPI (100 µg/mL stock) and Vectashield (Vector Laboratories, Burlingame, CA, USA). Five to 10 well-spread metaphase chromosomes were chosen and were observed under an Olympus BX53 fluorescence microscope with a built-in CCD camera (CoolSNAP ™ cf), using an oil lens (×100 magnification). One metaphase chromosome spread was selected as a representative for each genome for karyotype analyse. Karyograms were finalized using Adobe Photoshop CC version 25.3, whereas ideograms were generated using Adobe Illustrator CC version 22.4. Homologous chromosomes were identified based on their FISH signals, morphological characteristics, and lengths, considering previous karyotype data for *B. rapa*, *B. oleracea*, and *B. napus* [5], *B. nigra* and *B. juncea* [42], and *R. sativus* and ×*Brassicoraphanus* [43].

## 3. Results

### 3.1. Identification of Major Microsatellites in the A, C, B, and R Genomes

To capture microsatellites with higher-order organization, we loosen our definition of microsatellites from a stricter 2–5 bp [44] to an extended target length of ≤ 20 bp (Appendix A). This configuration generated 467 microsatellites in both strand orientations, including redundant or imperfect motif sequences [45] as demonstrated by sequence clustering (Figure 1A). Among the 467 microsatellites, the A, C, and B genomes had 123 sequences each. However, the R genome had only 98. The total base count for all microsatellites in each genome represented up to 0.2–0.3% of each genome (Figure 1B). Clustering of the related sequences generated 184 unique consensus sequences. Most of these microsatellites were di- and trinucleotides (Figure 1C). Many of these sequences represented <6 kb of cumulative total bases from the A, C, B, and R genomes. We only selected microsatellites with >6 kb cumulative total bases for FISH validation because microsatellites with lower total bases may not generate detectable FISH signals. This filtering retained 22 microsatellites, of which ACBR_msat01 had the highest cumulative total bases of ~328 kb (Figure 1D), which was distributed proportionately among the four genomes (Figure 1E and Table 3). Moreover, the ranks of the 22 microsatellites varied in each genome. For example, although the ACBR_msat06 was ranked sixth based on cumulative abundance, it ranked third, ninth, ninth, and fourteenth in the C, A, R, and B genomes, respectively.

The lengths of the 22 microsatellites ranged from 2 to 17 bp. However, the most abundant microsatellites were di- and trinucleotides. Moreover, 7-bp microsatellites were also abundant, although significantly biased to the C genome (Figure 1C). The 7-bp microsatellites represented the canonical *Arabidopsis*-type plant telomere repeat, ACBR_msat06, and its variant, ACBR_msat20 (Table 3). The *Arabidopsis*-type telomeric repeat covered ~65 kb total bases in the C genome, whereas only <15 kb in the other genomes. The ACBR_msat20, which was also observed predominantly in the C genome, had the central “A” in the *Arabidopsis*-telomeric repeat sequence replaced with a “C”.

Furthermore, microsatellites comprising only T and A were predominantly abundant in the C genome (Table 3). Although this bias was observed in di-, tri-, tetra, and pentanucleotides, it was prominent in the TA dinucleotide microsatellite (ACBR_msat02), in which the C genome had a total base of ~180 kb, whereas the A and B genomes had only < 20 kb and the R genome was ~71 kb (Table 3). The biased abundance of several microsatellites toward the C genome explains the higher proportion of microsatellites in the C genome than in the other three genomes (Figure 1B).

Similarly, ACBR_msat14 showed a biased abundance in the A genome (Figure 1E and Table 3). ACBR_msat14 is a 17-bp higher-order organized *Arabidopsis*-type telomere repeat composed of two units and an insertion of a trinucleotide “AGG” in the second repeat unit, particularly between the second and third T nucleotides (Table 3).

Some microsatellites, such as ACBR_msat18 and ACBR_msat22, also showed biased abundance in the B and R genomes, respectively. However, whether these genome-specific biases in microsatellite abundance also reflect their usability as genome-specific cytogenomic markers should be verified by FISH.

### 3.2. FISH Revealed Different Chromosome and Genome Distribution Patterns of the Microsatellites

To visualize the chromosomal distribution of the 22 microsatellites and analyze whether the microsatellites that showed putative genome specificity based on the in-silico analysis could also generate genome-specific FISH signals, we performed the FISH analysis in the four diploid A, C, B, and R genomes using PLOPs developed from the 22 microsatellite sequences (Appendix A).

Eleven of the 22 microsatellites did not show detectable FISH signals, indicating non-clustering of the microsatellite loci [46], making them undetectable by FISH (Appendix A). Other microsatellites showed clustered loci that generated detectable FISH signals in either a few or all chromosomes in each genome. One microsatellite showed signals in the A, C, and B genomes, whereas five showed non-specific signals for all four genomes. Two microsatellites showed signals in both the C and R genomes, and one microsatellite showed specific signals to the A (ACBR_msat14) and C (ACBR_msat20) genomes (Figure 2 and Figure 3, Table 3). We did not observe any microsatellite that discriminated the B and R genomes. However, we identified one microsatellite that showed signals to both B and R genomes (ACBR_msat01) (Appendix A, Figure 2 and Figure 3, Appendix A).

Genome-specific abundance from the in-silico analysis was partially supported by FISH data. The *Arabidopsis*-type telomeric repeat showed biased intense signals in the C genome, corroborating in silico data (Appendix A). In addition, ACBR_msat14 and ACBR_msat20 showed A and C genome-specific signals, which also corroborated the in-silico data (Appendix A, Figure 2 and Figure 3, Appendix A). However, ACBR_msat01 showed signals only in the B and R genomes as well as in one chromosome pair in the A genome. This did not coincide with the in-silico data, where relatively equal abundance in the four genomes was observed Figure 1E, Figure 2, and Figure 3, Table 3). These data suggest differences in the loci organization of this microsatellite in the four genomes: clustered in the B and R genomes as well as in chromosome 8 in the A genome and non-clustered in the C genome and other chromosomes in the A genome.

In summary, the FISH analysis revealed three microsatellites specific to the diploid species in the U’s triangle: ACBR_msat14 for the A genome, ACBR_msat20 for the C genome, and ACBR_msat01 for the B genome (Appendix A). Additionally, ACBR_msat01 could also detect R-genome chromosomes, making it a candidate marker for identifying the R chromosomes in the intergeneric ×*Brassicoraphanus* hybrids.

### 3.3. Genome-Specific Microsatellites Discriminated Subgenomes in Allotetraploids

To verify whether the FISH signals of ACBR_msat01, ACBR_msat14, and ACBR_msat20 in the diploid *Brassica* genomes were genome-specific and could distinguish subgenomes in interspecific and intergeneric hybrids, we performed the FISH analysis in allotetraploids within the U’s triangle and ×*Brassicoraphanus*.

As expected, the 10, 9, and 8 homologous chromosomes of the A, C, and B genomes were clearly distinguished from each other in *B. napus* (AACC), *B. juncea* (AABB), and *B. carinata* (BBCC) using ACBR_msat14, ACBR_msat20, and ACBR_msat01, respectively (Figure 4, Figure 5 and Figure 6). The *Arabidopsis*-type telomeric repeat also enabled identification of the C-genome chromosomes in *B. napus* and *B. carinata* (Appendix A), making this probe sufficient to distinguish the C-genome chromosomes among the allotetraploids in the U’s triangle.

Moreover, the A and R genomes were also clearly distinguished in ×*Brassicoraphanus* “BB#1” using ACBR_msat14 and ACBR_msat01, respectively (Figure 4 and Figure 5). However, although the R genome showed intense ACBR_msat01 signals, A-genome chromosomes also showed ACBR_msat01 signals in intercalary regions of all chromosomes with varied intensity, although weaker than those in the R genome, except for chromosome 8, which showed a more intense single locus at the proximal region of the long arm (Figure 5).

In summary, the three microsatellites proved to be efficient FISH probes to distinguish between the A, C, and B subgenomes of the allotetraploid species in the U’s triangle and between the A and R genomes in ×*Brassicoraphanus* (Figure 6).

### 3.4. FISH Revealed Genome Rearrangement and Chromatin Elimination in ×Brassicoraphanus 977

No cytogenetic analysis has been performed to investigate the chromosomal constitution of the putative tri-genome ×*Brassicoraphanus* 977. Therefore, we used the genome-specific microsatellites identified in this study to analyze the genomic composition of this plant. Line 977 was generated from a cross between two ×*Brassicoraphanus* allotetraploids, BB#50 (AARR, 2*n* = 4x = 38) and ‘‘Mooyangchae’’ (CCRR, 2*n* = 4x = 36). Therefore, we expected to see the complete set of the R chromosomes and a mixture of A and C chromosomes.

As expected, ×*Brassicoraphanus* 977 had 2*n* = 38 chromosomes, and ACBR_msat01 identified the complete R-genome chromosomes (Figure 5). ACBR_msat14 generated signals in the remaining 10 homologous pairs, indicating the A-genome origin of the other chromosomes. ACBR_msat20 generated FISH signals in three A-genome chromosomes, which we named A1C, A3C, and A6C, indicating introgression of C genome chromosome blocks into these chromosomes (Figure 5 and Figure 6). These data also suggest that the C genome chromatin is preferentially eliminated in ×*Brassicoraphanus* 977. The ACBR_msat20 signals were coincidentally localized in the 45S rDNA-bearing chromosomes and co-localized with the NOR signals. This observation indicates a role for 45S rDNA in the genome reshuffling of the A and C genomes and the homing of the ACBR_msat20 in the 45S rDNA array in ×*Brassicoraphanus* 977.

In summary, the FISH data of the genome-specific microsatellite probes in ×*Brassicoraphanus* 977 suggests the possibility that line 977 has an AARR genomic background with introgressed C-genome chromatin at the 45S rDNA loci, contrary to ARRC that was initially thought. This analysis also showed preferential elimination of the C genome in ×*Brassicoraphanus* 977.

### 3.5. Ribosomal DNA and Telomeric Repeat Distribution

We used 5S and 45S rDNA probes to identify known chromosomes and the *Arabidopsis*-type telomeric repeats to identify chromosome ends for karyotyping. All four diploid species revealed three 45S rDNA loci, whereas *B. rapa*, *B. nigra*, *B. oleracea*, and *R. sativus* had three, one, one, and two 5S rDNA loci, respectively, similar to those observed in previous studies [5,47,48,49]. In the allotetraploid species, *B. juncea*, *B. napus*, *B. carinata*, ×*Brassicoraphanus* ‘‘BB#1′’, and 977 had four, six, two, five, and six 5S rDNA loci and six, seven, six, five, and seven 45S rDNA loci. *Arabidopsis*-type telomeric repeat sequences were detected in all chromosomes of *Brassica* and *Raphanus* genome.

## 4. Discussion

### 4.1. Importance of Subgenome Discrimination in Allopolyploids

The ability to discriminate individual subgenomes or chromosomes in allopolyploids is invaluable for studying genome structure, dynamics, and stability [26,50]. This is particularly useful in analyzing the chromosomal composition and genetic stability of synthetic plants from intergeneric crosses, such as ×*Brassicoraphanus*, which generate sterile progenies from unstable meiotic chromosome paring [51].

Although GISH has been the conventional choice to discriminate subgenomes in allopolyploids [20,21], recent innovations, such as bulk oligo-FISH which uses single-copy oligomer probes, can allow easy discrimination of subgenomes and individual chromosomes [52]. However, this method requires high-quality genome assembly to design oligoprobe libraries, making it difficult to analyze species without a whole-genome assembly [52,53]. Moreover, the synthesis of bulk oligo libraries is costly relative to repeat-based pre-labeled oligomer probes (PLOPs) [38]. However, several bioinformatics tools (see Methods, for example) offer ways to analyze repeats and microsatellites from short next-generation sequences, enabling microsatellite analysis in species without genome assembly.

Microsatellites can have genome-specific distribution. Therefore, they can be useful in discriminating subgenomes in allopolyploids, as demonstrated in previous studies [25,26,29,54] and this study. Here, we identified 22 high-abundance microsatellites from the A, C, and B genomes of *Brassica* and the R genome of *R. sativus* using short next-generation sequencing reads. Three microsatellites, namely ACBR_14, ACBR_20, and ACBR_01, could discriminate the A, C, and B *Brassica* genomes and the *R. sativus* R genome, respectively, making them useful cytogenomic markers for natural and synthetic allopolyploids carrying the A, C, B, and R genomes.

Moreover, the FISH procedure using microsatellites is more straightforward and not as laborious and resource-intensive as GISH because PLOPs are used instead of genomic and blocking DNA. In addition, the use of rDNA and telomere repeat PLOPs and the new microsatellites enabled efficient homologous chromosome identification in each subgenome.

### 4.2. Organization of Microsatellite in the Chromosomes Determine FISH Detectability

The abundance of microsatellites in a genome cannot always predict detectability using FISH. Microsatellites are organized either in a clustered or non-clustered manner, making them detectable or undetectable by FISH, respectively [26,46,55,56,57]. Long microsatellite arrays or short arrays clustered together can allow amplified FISH signals to reach the fluorescence detection threshold [46].

This difference in locus organization was well demonstrated in ACBR_msat01. Intense ACBR_msat01 FISH signals were predominantly observed in the B and R genomes, despite being present in comparable abundance in all four genomes based on in silico quantification (Figure 1 and Table 3). This pattern suggests a more clustered organization of ACBR_msat01 in the B and R genomes compared to a predominantly non-clustered organization in the C and A genomes, except for one cluster in chromosome A8.

### 4.3. Genome-Specific Microsatellites for the A, C, and B Brassica Genomes

Our data corroborated the abundance of a telomeric repeat-like C-genome-specific microsatellite (ACBR_msat20) previously identified in *B. napus* using conventional molecular techniques [25]. However, although we only identified the consensus TTTCGGG sequence of ACBR_msat20, a previous study presented an intermittent organization of TTTCGGG, TTTGGGG, and a few imperfect derivative sequences [25], demonstrating that molecular analyses are relevant in defining actual sequences. However, for the FISH and genome discrimination, our results were comparable to previously reported data for discriminating the C-genome chromosomes from the A and B *Brassica* genomes.

In addition to the C genome-specific microsatellite, we identified ACBR_msat14, which is an A genome-specific microsatellite that efficiently discriminated A from the C and B genomes. Although it is challenging to precisely determine whether ACBR_msat14 is located in the telomeric or subtelomeric region just by looking at the FISH image, ACBR_msat14 is possibly localized at the subtelomeric region rather than the telomeres because (i) *Arabidopsis*-type telomeric signals were distinctly observed at chromosomal ends and (ii) subtelomeres are known to host different telomeric sequence variants, similar to the six different variants found in rice subtelomeres [58,59]. Subtelomeres are also prone to accumulation and rapid expansion of several tandem repeats, making subtelomere sites of frequent chromosomal rearrangements [60,61].

Similar to the C and A genomes, we also identified ACBR_msat01, which discriminated B from the A or C *Brassica* genomes in the U’s triangle. However, ACBR_msat01 failed to discriminate B from the R genome of *R. sativus*. In contrast, ACBR_msat01 can be an excellent cytogenetic marker between the species in the U’s triangle, it would not be useful to distinguish B and R chromosomes in allopolyploids comprising these two genomes. In this case, *B. nigra* and *R. sativus* centromeric satellites may be used to identify the B and R genomes, respectively [62,63,64]. However, no comparative analysis has been performed in allopolyploids with the B and R genomes that show genome specificity and a lack of cross-hybridization of these centromeric satellites.

### 4.4. Microsatellite Distribution Supports Phylogenetic Relationships of the Four Diploid Species

The distribution of the three genome-specific microsatellites reflected the phylogenetic relationships of the diploid species. Clustering of the ACBR_msat01 in the B and R genomes, but not in the A and C genomes, corroborates the observed closer relationship between *B. nigra* and *R. sativus* than *B. nigra* is to the *B. rapa* or *B. oleracea* [37]. Considering the high evolvability of microsatellites [65], the relatively recent divergence of the *B. rapa* and *B. oleracea* genomes from the more primitive *B. nigra* and *R. sativus* [5,37] may have abruptly de-clustered the ACBR_msat01 loci in the C genome and most of the A genome, except for that chromosome A8. This genome reshuffling may have also concomitantly spurred the differential evolution of telomere-like repeats between the A and C genomes.

### 4.5. Association of 45S rDNA in Genome Reshuffling

Several studies have implicated the involvement of the 45S rDNA array in plant genome rearrangements [60,66,67,68]. Similarly, we observed the participation of the 45S rDNA array in the rapid evolution of the ACBR_msat20 microsatellite in ×*Brassicoraphanus* 977. While the ACBR_msat20 microsatellite was dispersed in the C-genome chromosomes of the diploid and allopolyploid species in the U’s triangle, it was localized at the 45S rDNA loci in ×*Brassicoraphanus* 977. This observation further demonstrates the role of 45S rDNA in genome reshuffling. Although we have not characterized the sequence of the 45S rDNA in ×*Brassicoraphanus* 977, the ACBR_msat20 microsatellite must have been likely inserted into the 45S rDNA intergenic spacer, as this region is often considered a “logistics hub” of repeats in the genome, hosting different types of repeats into or out of the 45S rDNA intergenic spacer [68,69,70]. The 45S rDNA may also facilitate the rapid and efficient concerted expansion and contraction of genomic repeats, as observed in *Senna* [68] and the ACBR_msat20 microsatellite in ×*Brassicoraphanus* 977.

Although the exact mechanism by which 45S rDNA interacts with the genome and performs genome rearrangements is not yet fully understood, microhomology between sub-repeats in the 45S rDNA intergenic spacer and microsatellites may drive recombination, particularly given that microsatellites are known hotspots for recombination and chromosomal rearrangements [71].

## 5. Conclusions

We performed a comprehensive comparative microsatellite quantification and FISH analysis between the *Brassica* A, C, B, and *R. sativus* R genomes and identified three microsatellites that could discriminate the A, C, and B/R genomes. Along with the 45S and 5S rDNA and the *Arabidopsis*-type telomeric repeats, we efficiently identified the subgenomes and homologous chromosomes of the natural and synthetic allopolyploids between these four genomes. This study presents the first three-genome integrated karyotype of the ×*Brassicoraphanus* 977. The genome-specific microsatellites studied here could constitute an excellent cytogenomic marker to study several lines of synthetic ×*Brassicoraphanus.*

Although the ACBR_msat01 could not distinguish between the B and R genomes, further comparative analysis with longer satellite sequences may generate genome-specific probes, such as the centromeric repeats identified in the B and R genomes [62,63,64]. However, further FISH validation is needed to determine whether these centromeric repeats cross hybridize with each other, considering the close genetic distance between the B and R genomes.

We have also shown the rapid evolution of the ACBR_msat20 microsatellite and implied the involvement of 45S rDNA in genome reshuffling. Moreover, further sequence characterization of the complete 45S rDNA sequence, including the intergenic spacer in ×*Brassicoraphanus* 977, will explain the observed signals of the ACBR_msat20 in the 45S rDNA array and reveal its actual insertion site. This analysis will also help provide clues on the role and mechanism of 45S rDNA in genome rearrangements.

## Figures and Tables

**Figure 1 cells-10-02358-f001:**
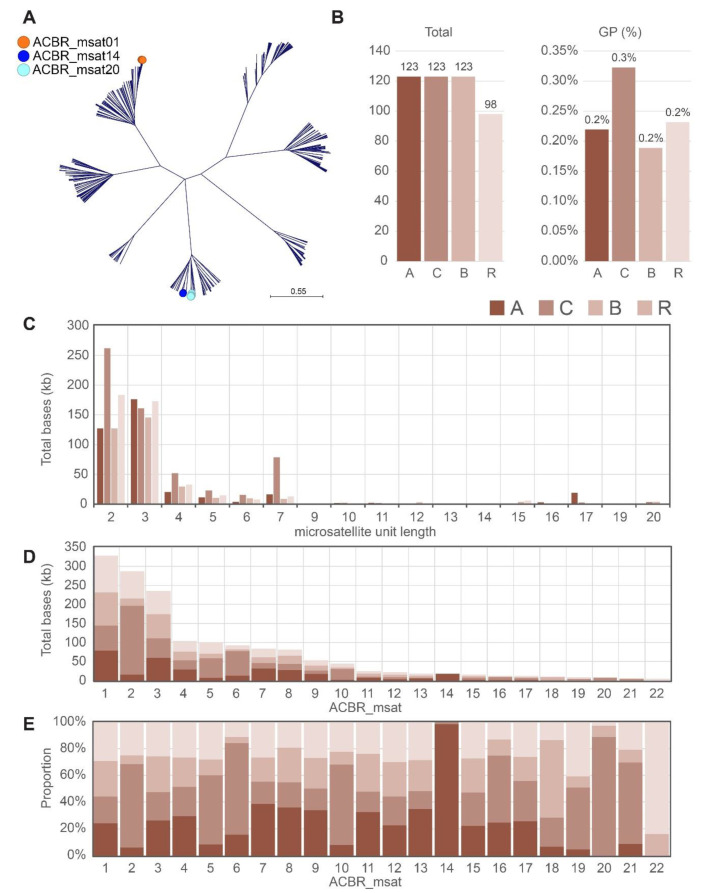
Quantification of microsatellites in the A, C, B, and R genomes. (**A**) Clustering of 467 microsatellites that were ≤ 20 bp in the A, B, C, and R genomes. The three microsatellites that showed genome-specific distribution are shown in circles. (**B**) Distribution of the 467 microsatellites by genome in terms of total counts and genome proportions (GP). (**C**) Abundance of microsatellites according to monomer length. (**D**) Cumulative base count of the 22 microsatellites that had a cumulative base count of ≥ 6 kb. The numbers in the *x*-axis correspond to the microsatellite names listed in Table 3. (**E**) Cumulative proportion of each microsatellite in the A, C, B, and R genomes. Note the predominant abundance of ACBR_msat14 and ACBR_msat20 in the A and C genomes, respectively. ACBR_msat01 showed relative equal abundance in the four genomes. Bar graph colors in C–E are same as those in B.

**Figure 2 cells-10-02358-f002:**
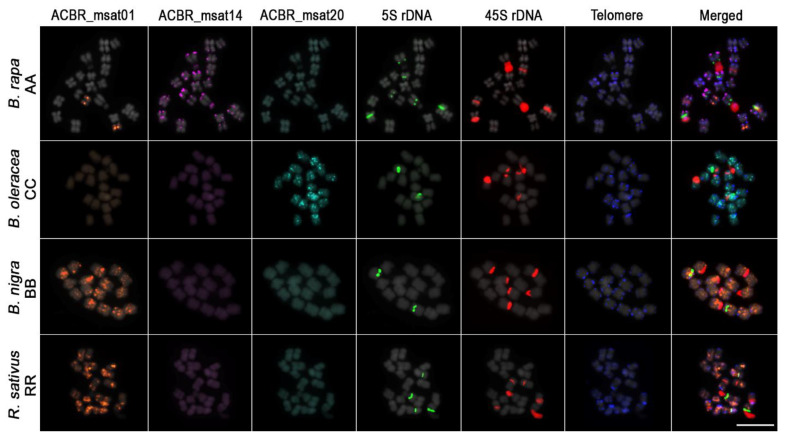
FISH analysis of ACBR_msat01, ACBR_msat14, and ACBR_msat20 in the four diploid genomes. ACBR_msat14, and ACBR_msat20 were exclusively detected in *B. rapa* and *B. oleracea* genomes, respectively. ACBR_msat01 was detected in both *B. nigra* and *R. sativus* genomes including one pair of *B. rapa* chromosomes. Scale bar = 10 µm.

**Figure 3 cells-10-02358-f003:**
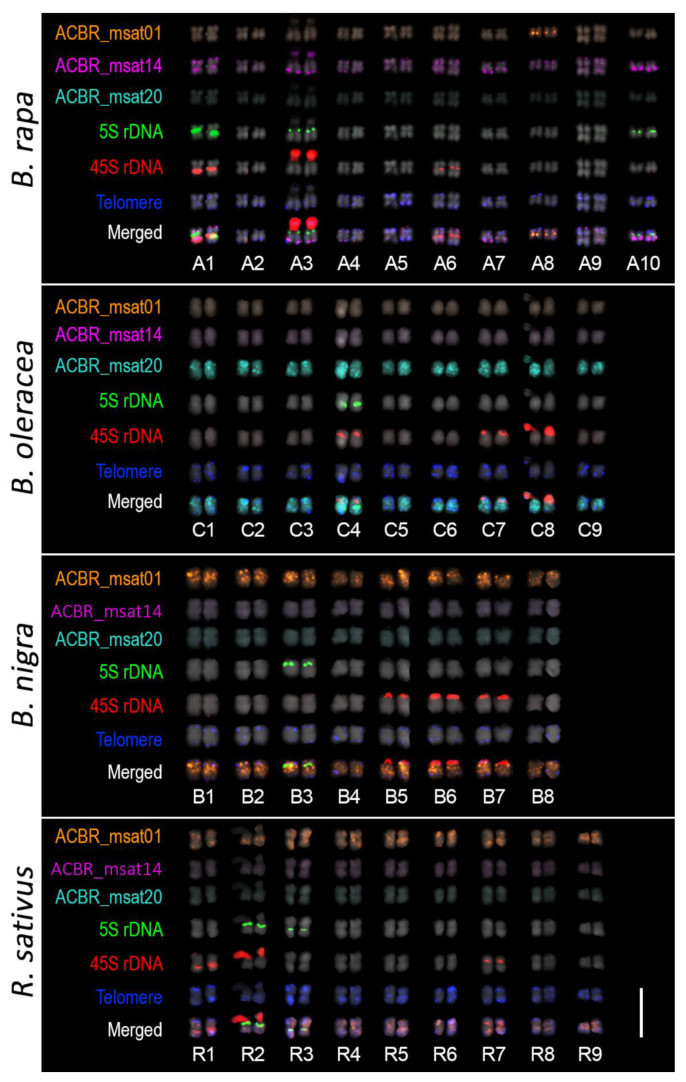
Karyogram of *B. rapa*, *B. oleracea*, *B. nigra*, and *R. sativus* chromosomes from Figure 2. The chromosomal distribution of ACBR_msat01, ACBR_msat14, ACBR_msat20, 5S rDNA, 45S rDNA and *Arabidopsis*-type telomere repeats as well as the merged images of the six probes are shown. Note the hybridization of ACBR_msat01 in chromosome A8. Also note the disperse distribution of ACBR_msat20 signal in the C genome. Scale bar = 10 µm.

**Figure 4 cells-10-02358-f004:**
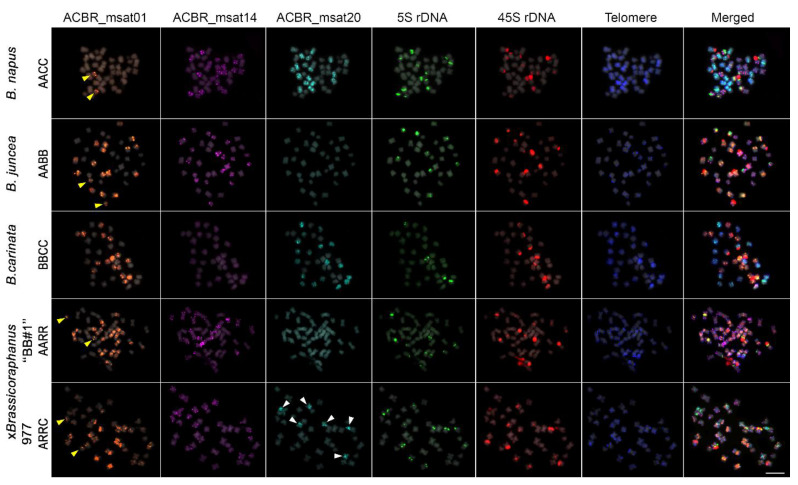
FISH analysis of ACBR_msat01, ACBR_msat14, and ACBR_msat20 in *B. napus*, *B. juncea*, *B. carinata*, ×*Brassicoraphanus* ‘BB#1′, and ×*Brassicoraphanus* 977. ACBR_msat14, and ACBR_msat20 clearly discriminated the A and C genomes in the allopolyploids, respectively. ACBR_msat01 was useful in discriminating the B-genome in the interspecific AABB and BBCC allopolyploids, the R genome in the intergeneric AARR genome, and chromosome A8 (yellow arrowheads). Note the six chromosomes with ACBR_msat20 in ×*Brassicoraphanus* 977 indicating C-genome chromosome blocks (white arrowheads). Scale bar = 10 µm.

**Figure 5 cells-10-02358-f005:**
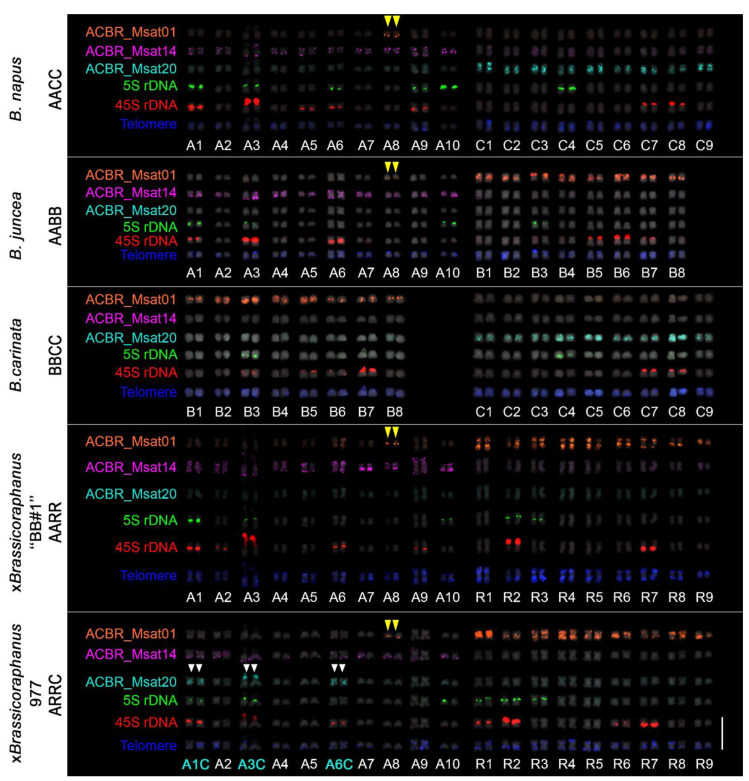
Karyogram of natural and synthetic *Brassica* and *Raphanus* allotetraploids. The chromosomal distribution of ACBR_msat01, ACBR_msat14, ACBR_msat20, 5S rDNA, 45S rDNA and *Arabidopsis*-type telomere repeats are shown. ACBR_msat14 localized at the subterminal with some in the interstitial regions in the A-genome chromosomes (A1–A10). ACBR_msat20 localized at the interstitial regions of C-genome chromosomes (C1–C9). ACBR_msat01 localized at the interstitial regions of B-genome chromosomes (B1–B8) and extra signals at the paracentric region of chromosome A8 (yellow arrowheads). A-genome chromosomes with ACBR_msat20 signals are shown in white arrowheads. Scale bar = 10 μm.

**Figure 6 cells-10-02358-f006:**
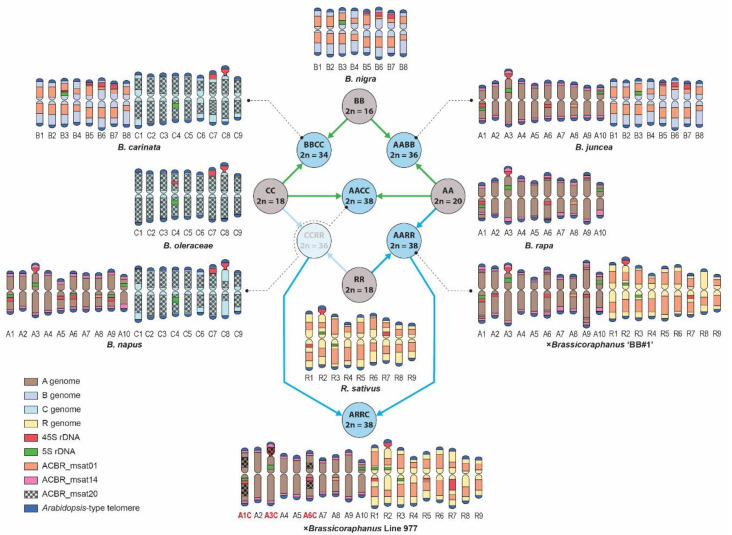
Karyotype ideograms and genome relationships between the four diploids and their allotetraploid hybrids. Green arrows show the genome relationships among species within the U’s Triangle, and blue arrows for those involving the R genome. The A, C, B, and R genomes are distinguished by chromosome color. The chromosome numbers of ×*Brassicoraphanus* 977 with ACBR_msat20 signals are highlighted red.

**Table 1 cells-10-02358-t001:** Description of *Brassica* and *Raphanus* next-generation sequencing reads used to generate the microsatellites and summary of the coverage.

Species	Cultivar	Genome	Genome Size (Mbp)	Accession Number	Reads	Coverage (×)
*B. rapa*	Chiifu-401-42	AA	353	GCA_000309985.3	199,662	0.06
*B. oleracea*	TO1000	CC	522	GCA_000695525.1	200,880	0.04
*B. nigra*	inbred line YZ12151	BB	468	GCA_001682895.1	201,798	0.04
*R. sativus*	XYB36-2	RR	414	GCA_002197605.1	201,210	0.05

**Table 2 cells-10-02358-t002:** List of plant materials with their corresponding haploid chromosome number, genome type, seed source, and accession number.

Species	2*n*	Genome Type	Source	Accession Number
*B. rapa*	20	AA	NAAS^a^	IT 032730
*B. oleracea*	18	CC	NPGS^b^	PI 24501510GI
*B. nigra*	16	BB	NPGS	PI 649154
*R. sativus*	18	RR	danong.co.kr (accessed on 31 April 2019)	N/A
*B. napus*	38	AACC	NPGS	TI 031006
*B. juncea*	36	AABB	NPGS	PI 633077
*B. carinata*	34	BBCC	NPGS	Ames 2779
*×Brassicoraphanus* ‘BB#1′	38	AARR	BioBreeding Institute (Ansung, Korea)	PRJNA353741
*×Brassicoraphanus* 977	38	ARRC	BioBreeding Institute	N/A

^a^ National Academy of Agricultural Sciences, (Jeollabuk-do, Korea); ^b^ National Plant Germplasm System, (US Department of Agriculture, USA).

**Table 3 cells-10-02358-t003:** Summary of top 22 microsatellites identified in the A, C, B, and R genomes.

Name	Sequence	Length	Total Bases				
			A	C	B	R	Grand Total
ACBR_msat01	AG	2	80,004	65,645	87,523	95,545	328,717
ACBR_msat02	TA	2	17,429	180,550	18,852	71,783	288,614
ACBR_msat03	AAG	3	62,101	50,474	63,611	61,099	237,285
ACBR_msat04	TGA	3	31,243	23,210	23,246	28,005	105,704
ACBR_msat05	TAA	3	8541	51,851	11,968	28,179	100,539
ACBR_msat06	TTTAGGG	7	14,629	64,754	4438	10,471	94,292
ACBR_msat07	TTG	3	33,032	14,323	15,292	22,713	85,360
ACBR_msat08	TG	2	29,815	15,745	21,198	16,033	82,791
ACBR_msat09	GAG	3	18,862	8880	12,870	14,823	55,435
ACBR_msat10	AAAT	4	3720	27,736	4390	10,228	46,074
ACBR_msat11	ACC	3	8494	4084	7375	6254	26,207
ACBR_msat12	CTTT	4	5361	5163	6096	7137	23,757
ACBR_msat13	TGC	3	6967	2710	4611	5726	20,014
ACBR_msat14	TTTAGGGTTAGGTAGGG	17	18,903	297	0	0	19,200
ACBR_msat15	AAAC	4	3933	4403	4519	4813	17,668
ACBR_msat16	TTCGG	5	3505	7088	1709	1843	14,145
ACBR_msat17	ACT	3	3484	4086	2441	3552	13,563
ACBR_msat18	ATAG	4	850	2701	7258	1684	12,493
ACBR_msat19	ATTTT	5	509	4749	866	4191	10,315
ACBR_msat20	TTTCGGG	7	0	8845	835	270	9950
ACBR_msat21	AATT	4	722	4925	792	1661	8100
ACBR_msat22	TGAACAGTGTTTCGA	15	0	0	973	5126	6099

## Data Availability

All data that support the findings of this study are presented in this article and the Appendix A.

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
