# Peer review of "Subgenome Discrimination in *Brassica* and *Raphanus* Allopolyploids Using Microsatellites"

_cells, 2021, doi:10.3390/cells10092358_

Round 1
Reviewer 1 Report
A bit more on the use of microsatellites (and their alleles) to discriminate among the subgenomes (i.e. monoploid genomes) of allopolyploids in general would help provide a better background in the Introduction. Prior studies show that microsatellites may work across many species, but not all. The authors can then state that they are taking the next step, by using the presence and absence of individual msats among the component chromosomes of subgenomes (i.e. monoploid genomes) to help establish which chromosomes are homologous in an allopolyploid and the subgenome they are associated with.
A solid paper to consider is:
Peakall R, Gilmore S, Keys W, Morgante M, Rafalski A. 1998. Cross-species amplification of soybean (Glycine max) simple sequence repeats (SSRs) within the genus and other legume genera: implications for the transferability of SSRs in plants. Mol Biol Evol. 15(10):1275-87. doi: 10.1093/oxfordjournals.molbev.a025856. PMID: 9787434.
Also, msat alleles have been used to discriminate among the subgenomes (i.e. monoloid genomes) in gametophytically allotriploid peat mosses which are intersubgeneric allopolyploids (Karlin et al. 2009, 2011). Thus it was possible to discriminate among subgenomes based on msat alleles, but not to discriminate among the component chromosomes. Their studies further found that some msats did not work across all of the ancestral subgenomes.
Karlin, E. F., S. B. Boles, M. Ricca, E. Temsch, J. Greilhuber & A. J. Shaw. 2009. Three-genome mosses: complex double allopolyploid origins for triploid gametophytes in Sphagnum. Molecular Ecology 18: 1439–1454.
Karlin, E. F., S. B. Boles, R. D. Seppelt, S. Terracciano & A. J. Shaw. 2011. The peat moss Sphagnum cuspidatum in Australia: microsatellites provide a global perspective. Systematic Botany 26: 22–32.
Author Response
Reviewer 1:
A bit more on the use of microsatellites (and their alleles) to discriminate among the subgenomes (i.e. monoploid genomes) of allopolyploids in general would help provide a better background in the Introduction. Prior studies show that microsatellites may work across many species, but not all. The authors can then state that they are taking the next step, by using the presence and absence of individual msats among the component chromosomes of subgenomes (i.e. monoploid genomes) to help establish which chromosomes are homologous in an allopolyploid and the subgenome they are associated with.
A solid paper to consider is:
Peakall R, Gilmore S, Keys W, Morgante M, Rafalski A. 1998. Cross-species amplification of soybean (Glycine max) simple sequence repeats (SSRs) within the genus and other legume genera: implications for the transferability of SSRs in plants. Mol Biol Evol. 15(10):1275-87. doi: 10.1093/oxfordjournals.molbev.a025856. PMID: 9787434.
Also, msat alleles have been used to discriminate among the subgenomes (i.e. monoloid genomes) in gametophytically allotriploid peat mosses which are intersubgeneric allopolyploids (Karlin et al. 2009, 2011). Thus it was possible to discriminate among subgenomes based on msat alleles, but not to discriminate among the component chromosomes. Their studies further found that some msats did not work across all of the ancestral subgenomes.
Karlin, E. F., S. B. Boles, M. Ricca, E. Temsch, J. Greilhuber & A. J. Shaw. 2009. Three-genome mosses: complex double allopolyploid origins for triploid gametophytes in Sphagnum. Molecular Ecology 18: 1439–1454.
Karlin, E. F., S. B. Boles, R. D. Seppelt, S. Terracciano & A. J. Shaw. 2011. The peat moss Sphagnum cuspidatum in Australia: microsatellites provide a global perspective. Systematic Botany 26: 22–32.
Answer: We have addressed these suggestions in the main text.
Reviewer 2 Report
The manuscript entitled “Genome-Specific Microsatellites for Subgenome Discrimination in Brassica and Raphanus Allopolyploids” aimed to identify microsatellites as probes in identifying subgenomes within closely related Brassica and Raphanus species. The authors quantified microsatellite repeats in the A, C, B, and R genomes (of Brassica and Raphanus species) and performed a comparative FISH to identify potential genome-specific microsatellites that can be used as a cytogenetic resource in karyotyping polyploid hybrids between these four genomes. The research idea is very important, and the findings provide a genetic useful tool for studying the cytogenetic stability in allopolyploids. Generally, the manuscript is very good and well-written however, the main problem in the manuscript is lacking the statistical analyses. The authors did not mention any statistical analysis in the methods section as I can see a reflection of any statistical analysis on the figures. In addition, most of the references are too old and need to be updated.
Minor changes:
- The range of the Y-axis in Figure 1C should be increased as the maximum range is 250 however, the bar (2 mer) is above this range.
- The titles of axes in Fig 1 are confusing. For example, X-axis tiles are not complete and Y-axis in Figure 1C and D are confusing (both are “total bases”). Please edit the titles to be informative and add statistical information and symbols.
- The following review for Brassica genomes is new and relevant to the topic of the manuscript, please cite. Ebeed, H. T. (2020). Bioinformatics Studies on the Identification of New Players and Candidate Genes to Improve Brassica Response to Abiotic Stress. In The Plant Family Brassicaceae (pp. 483-496). Springer, Singapore.
Author Response
Answers to reviewers’ comments:
Reviewer 2:
The manuscript entitled “Genome-Specific Microsatellites for Subgenome Discrimination in Brassica and Raphanus Allopolyploids” aimed to identify microsatellites as probes in identifying subgenomes within closely related Brassica and Raphanus species. The authors quantified microsatellite repeats in the A, C, B, and R genomes (of Brassica and Raphanus species) and performed a comparative FISH to identify potential genome-specific microsatellites that can be used as a cytogenetic resource in karyotyping polyploid hybrids between these four genomes. The research idea is very important, and the findings provide a genetic useful tool for studying the cytogenetic stability in allopolyploids. Generally, the manuscript is very good and well-written however, the main problem in the manuscript is lacking the statistical analyses. The authors did not mention any statistical analysis in the methods section as I can see a reflection of any statistical analysis on the figures. In addition, most of the references are too old and need to be updated.
Minor changes:
- The range of the Y-axis in Figure 1C should be increased as the maximum range is 250 however, the bar (2 mer) is above this range.
Answer: This has been revised. - The titles of axes in Fig 1 are confusing. For example, X-axis tiles are not complete and Y-axis in Figure 1C and D are confusing (both are “total bases”). Please edit the titles to be informative and add statistical information and symbols.
Answer: The x-axis title in C have been revised. Clarification about the axes name in D is added in the figure legend. The y-axes in C and D all correspond to total bases. Panel C is based on unit length whereas panel D is based on the cumulative abundance in the A,C,B, and R genomes of each repeat sequence. - The following review for Brassica genomes is new and relevant to the topic of the manuscript, please cite. Ebeed, H. T. (2020). Bioinformatics Studies on the Identification of New Players and Candidate Genes to Improve Brassica Response to Abiotic Stress. In The Plant Family Brassicaceae (pp. 483-496). Springer, Singapore
Answer: We have added this reference.
Reviewer 3 Report
Paper is related to use of microsatellites to discriminate between economical important plant species. Please see below some observations made to improve the manuscript.
-Title as it is seems to be contradictory, if we have specific microsatellites for genome are those useful for subgenome given the fact that it would be some changes within the subgenome. Please rewrite.
-In the abstract is not clear what authors want to translate to audience, for instance why is necessary to discriminate donor genomes, please clarify.
-Conclusion part in the abstract is superfluous, authors need to finish with the utility of the research in this case, as it is, is superficial (L26-28).
-Please be consistent with the use of scientific names (see L14 vs L19, L14 vs L41-44), through the manuscript.
-Fig S1, S2. Please add more information to have a better reading in the figure.
-L73-81. Please re-write the objectives, so we have a single paragraph with all objectives.
-Additional information is needed in Table 1, 2, S1 for a better reading.
-Section 2.3, 2.4, and 2.45 is not clear how many replicates were used in the experiment. Or how many seeds were further processed for the experiments described.
-L132. Authors need to have consistency in given all the information of suppliers.
-L177. Based on what authors make such assumption (although...)?
-L336. Please modify repetitive sentences.
Author Response
Answers to reviewers’ comments:
Reviewer 3:
Paper is related to use of microsatellites to discriminate between economical important plant species. Please see below some observations made to improve the manuscript.
-Title as it is seems to be contradictory, if we have specific microsatellites for genome are those useful for subgenome given the fact that it would be some changes within the subgenome. Please rewrite.
Answer: The title has been updated.
-In the abstract is not clear what authors want to translate to audience, for instance why is necessary to discriminate donor genomes, please clarify.
Answer: This sentence has been revised.
-Conclusion part in the abstract is superfluous, authors need to finish with the utility of the research in this case, as it is, is superficial (L26-28).
Answer: The conclusion of the abstract has been updated.
-Please be consistent with the use of scientific names (see L14 vs L19, L14 vs L41-44), through the manuscript.
Answer: We have addressed this in the main text.
-Fig S1, S2. Please add more information to have a better reading in the figure.
Answer: We have addressed this suggestion in the main text.
-L73-81. Please re-write the objectives, so we have a single paragraph with all objectives.
Answer: We have addressed this suggestion in the main text.
-Additional information is needed in Table 1, 2, S1 for a better reading.
Answer: We have addressed this suggestion in the main text.
-Section 2.3, 2.4, and 2.45 is not clear how many replicates were used in the experiment. Or how many seeds were further processed for the experiments described.
Answer: We have addressed this question in the main text.
-L132. Authors need to have consistency in given all the information of suppliers.
Answer: We have addressed this in the text.
-L177. Based on what authors make such assumption (although...)?
Answer: If you look at Table 3, the 7-bp microsatellites are considerably more abundant in the C genome than any other genomes.
-L336. Please modify repetitive sentences.
Answer: we have modified the sentence.